# Local occurrence and fast spread of B.1.1.7 lineage: A glimpse into Friuli Venezia Giulia

Catia Mio[1©], Chiara Dal Secco[1©], Stefania Marzinotto[2], Claudio Bruno[1], Santa Pimpo[1], Elena Betto[2], Martina Bertoni[2], Corrado Pipan[1,2], Emanuela Sozio[3], Carlo Tascini[1,3], Giuseppe Damante[1,2], Francesco Curcio[1,2]*

1 Department of Medicine (DAME), University of Udine, Udine, Italy, 2 Department of Laboratory Medicine, University Hospital of Udine, Udine, Italy, 3 Infectious Diseases Clinic, University Hospital of Udine, Udine, Italy

© These authors contributed equally to this work.
* francesco.curcio@uniud.it

## Abstract

In-depth study of the entire SARS-CoV-2 genome has uncovered many mutations, which have replaced the lineage that characterized the first wave of infections all around the world. In December 2020, the outbreak of variant of concern (VOC) 202012/01 (lineage B.1.1.7) in the United Kingdom defined a turning point during the pandemic, immediately posing a worldwide threat on the Covid-19 vaccination campaign. Here, we reported the evolution of B.1.1.7 lineage-related infections, analyzing samples collected from January 1st 2021, until April 15th 2021, in Friuli Venezia Giulia, a northeastern region of Italy. A cohort of 1508 nasopharyngeal swabs was analyzed by High Resolution Melting (HRM) and 479 randomly selected samples underwent Next Generation Sequencing analysis (NGS), uncovering a steady and continuous accumulation of B.1.1.7 lineage-related specimens, joined by sporadic cases of other known lineages (i.e. harboring the Spike glycoprotein p.E484K mutation). All the SARS-CoV-2 genome has been analyzed in order to highlight all the rare mutations that may eventually result in a new variant of interest. This work suggests that a thorough monitoring of the SARS-CoV-2 genome by NGS is essential to contain any new variant that could jeopardize all the efforts that have been made so far to resolve the emergence of the pandemic.

## Introduction

The emergence and continuous spread of SARS-CoV-2 viral infections has seriously tested public health infrastructures, and led to millions of infections and deaths worldwide [1]. Over a year into the pandemic, sequencing analysis has shown that SARS-CoV-2 variants are being selected as the ongoing uncontrolled spreading of the SARS-CoV-2. While many mutations represent neutral genetic drift [2], a subset may alter its pathogenic potential and possibly reduce the efficacy of drugs and of immunity elicited by current vaccines. Since isolated, complete viral genome sequences have been periodically uploaded into the Global Initiative on Sharing Avian Influenza Data (GISAID; https://www.gisaid.org) database. GISAID initiated a naming system based on large clades that have been identified through variations from the

**Competing interests:** The authors have declared that no competing interests exist.

reference genome. GISAID-deposited genomes have been analyzed in different open-source project among which NextStrain has gained a well-deserved popularity (www.nextstrain.org). It interactively visualizes the phylogeny of an emerging viral pathogen to better understand its dispersion dynamic and to improve outbreak responses. Furthermore, it named evolutionary stable lineages based on the year of emergence and a letter (e.g., Nextstrain Clade: 19A, 19B, 20A, 20B and so on) (https://nextstrain.org/ncov/gisaid/global?l=radial; version v2.29.1), despite some sub lineages have been identified with additional information (e.g., 20E/EU1) but with no systematic rule. Indeed, in mid-2020 Rambaut and colleagues have suggested a third dynamic nomenclature, based on evolutionary relationships and epidemiological relevance and an algorithm named PANGOLIN (Phylogenetic Assignment of Named Global Outbreak Lineages) has been proposed and used to assign lineages to FASTA sequences [3]. Recently, to assist with public discussions of variants, the World Health Organization (WHO) introduced a non-stigmatizing label for SARS-CoV-2 variants using letters of the Greek Alphabet to aid the non-scientific audience. SARS-CoV-2 nomenclature methods have been revised by Alm and colleagues [4] and have been further summarized in S1 Table.

As infections continue to spread, viral genome sequencing coupled to epidemiological surveillance became mandatory to actively monitor all the SARS-CoV-2 variants that are globally circulating and rapidly determine the ongoing virus evolution. Based on this premises, several studies highlighted that the main circulating lineage in Italy were B.1 and B.1.177, together with their sub-lineages, representing about 70% of cases in late 2020 [5–7]. Later on, several variants of concern (VOC), each represented by a constellation of specific mutations selected to enhance viral fitness, have emerged [8]. In December 2020, a new variant was identified in the United Kingdom, firstly named as VOC 202012/01, and representing the PANGO B.1.1.7 lineage. This variant has been recently named Alpha by the WHO classification. It has an estimated 43–90% higher transmissibility than pre-existing lineages, thus causing an increasing concerns on Covid-19 infection rate [9]. In the same period, the VOC 202012/02 variant was first isolated in South Africa, which is associated the B.1.351 lineage, newly named Beta. Finally, in January 2021 a third VOC has been firstly identified in Brazil and then in Japan, named VOC 202101/02, belonging to the P.1 lineage, newly named Gamma. All these lineages were concerning due to their increased transmissibility and putative disease severity.

In this study, we aimed to track and evaluate the diffusion of the different SARS-CoV-2 lineages in the northeastern part of the Italian peninsula, in a cohort of nasopharyngeal swabs collected between January and April 2021.

## Materials and methods

### Samples' collection and RNA extraction

SARS-CoV-2 samples were collected from all the nasopharyngeal swabs resulted positive to the Allplex COVID-19 assay on Seegene's automated system between January 1st, 2021 and April 15th, 2021, in the Department of Laboratory Medicine of the University Hospital of Udine (Italy). SARS-CoV-2 positive samples were randomly selected and tested with primers targeting the *E* gene, which has been previously demonstrated in our reports to be the most sensitive target [10, 11]. Samples with Ct value ≤ 28 were chosen for further analyses. For swab collection, transportation, and long-term storage UTM® tubes (COPAN Diagnostics) were used. Ethical approval was obtained from the Medical Research Ethics Committee of the Region Friuli Venezia Giulia, Italy (Consent CEUR-2020-Os-033).

Total nucleic acids were extracted from 500 μL UTM-medium on the QIAsymphony SP using the QIAsymphony DSP Virus/Pathogen Midi Kit (Qiagen), following manufacturer's instructions. Samples were eluted in 60μL AVE buffer and used as template for downstream analysis.

## High resolution melting (HRM)

High resolution melting was used for the qualitative detection of p.N501Y and p.E484K mutations in the SARS-CoV-2 genome. Briefly, extracted RNA was used as a template for the VirSNiP SARS B1351 assay on the LightCycler ® 480 Real-Time PCR System (Roche Diagnostics). Positive and negative control (NTC) were included in all experiments. 4 μL Roche master mix, 0.5 μL reagent mix (containing primers and probes), 5.5 μL PCR-grade water and 10 μL of template were mixed to a total volume of 20 μL. Amplification was performed following manufacturer's instruction. After normalization and temperature shift determination, the different melting curves were generated. p.E484K is associated with a shift from 52˚C to 57˚C while p.N501Y is associated with a shift from 60˚C to 65˚C. LightCycler® 480 Software was used to analyze all data.

## Library preparation and next-generation sequencing (NGS)

Samples were randomly selected from the 1508 specimens that underwent HRM screening. 479 barcoded libraries were prepared using the Ion AmpliSeq SARS-CoV-2 Research Panel (Thermo Fisher Scientific) and the Ion AmpliSeq Library Kit Plus (Thermo Fisher Scientific), following manufacturer's protocol. The panel consists of 237 amplicons that allows for the sequencing of 99% of the SARS-CoV-2 reference genome (NC_045512.2), covering from position 43 to position 29,842. All the reactions were performed in a Veriti™Dx 96-Well Thermal Cycler (Applied Biosystems™). Libraries were quantified with the Qubit dsDNA HS Assay kit (Life Technologies) and then diluted to 30 pM. Libraries were loaded into the Ion Chef™ Instrument (Thermo Fisher Scientific) for template enrichment and chip loading. Sequencing was performed with the Ion S5 GeneStudio Sequencer using the Ion 510 & Ion 520 & Ion 530 Kit-Chef and the Ion 530™ chip-kit (all Thermo Fisher Scientific).

## Data analysis and variant prioritization

After alignment, the following plugins were run on the Torrent Suite™ Server (Thermo Fisher Scientific): the SARS_CoV_2_variant caller, the SARS_CoV_2_CoverageAnalysis, the SARS_-CoV_2_annotateSnpEff and the generateConsensus. Sequences kept for further investigations were the ones possessing a mean depth of coverage ≥ 500 and a percentage of gaps ≤ 20% of the entire sequence. For variant calling, variants with a genotype quality (GQ) score ≥ 30, a coverage (FDP) ≥ 500 and a minimum alternate allele frequency of 70% (AF ≥ 70%) were kept for further investigations. Clade and lineages of 479 specimens were defined with the Nextclade v2.29.1 web application (https://clades.nextstrain.org/), the GISAID CoVsurver mutation App (https://www.gisaid.org/epiflu-applications/covsurver-mutations-app/), the pangolin web app developed by the Centre for Genomic Pathogen Surveillance–version v.3.1.11 - (https://pangolin.cog-uk.io/) and the Ultrafast Sample placement on Existing tRee (UShER) web tool by UCSC (https://hgwdev-angie.gi.ucsc.edu/cgi-bin/hgPhyloPlace). SNPs and INDELs frequencies were evaluated in the CovidMiner data portal (https://covid-miner.ifo.gov.it/app/home) and in the GISAID database (https://www.gisaid.org/).

FASTA sequences have been uploaded in the GISAID database. S2 Table enlists Accession IDs and virus names.

## Phylogenetic analysis

FASTA sequences were aligned with ClustalW. The Maximum Likelihood (Tamura-Nei method) method was used to generate a phylogenetic tree of the 479 sequences. Alignment and tree were computed with Molecular Evolutionary Genetics Analysis (MEGA) v.11.

## Statistics

Nonlinear regression was performed mimicking a dose-response plot. Best-fitting curve was super-imposed with least squares fit. All statistical analysis were performed with GraphPad Prism 6.0.

## Results

The University Hospital of Udine is the hub that serves the 134 cities that are part of the former province of Udine (Italy) and has tested approximately 262'144 specimens since the third wave of the pandemic in January 2021. We retrospectively analyzed 1508 specimens, tested positive between January 1st, 2021 and April 15th, 2021 to track the occurrence and the potential diffusion of the SARS-CoV-2 variants of concern (VOC) and the variants of interest (VUI), reported by the World Health Organization (WHO) since December 2020. The presence of p. N501Y and p.E484K Spike mutations was assessed by high resolution melting (HRM). We witnessed a fast and exponential growth in the percentage of samples possessing the p.N501Y mutation. The first positive samples were already evidenced in January 2021 (N = 3) with a steady increase in the weeks that followed. Fig 1 panel A-D graphically represents the p.N501Y spread during January and February in our territory. Indeed, 99.1% of p.N501Y positivity was then reached in early April. Table 1 and Fig 1 panel E summarize the percentage of p.N501Y positive samples over the selected analytical time points. Indeed, our data agree with the so called "one-month rule of the B.1.1.7 take-over", according to which the relative abundance of this lineage goes from 20 to 80% in about four weeks [12, 13].

Furthermore, samples were randomly selected and analyzed by next generation sequencing (NGS) to confirm which lineage they belonged to and to obtain the entire genomic sequence.

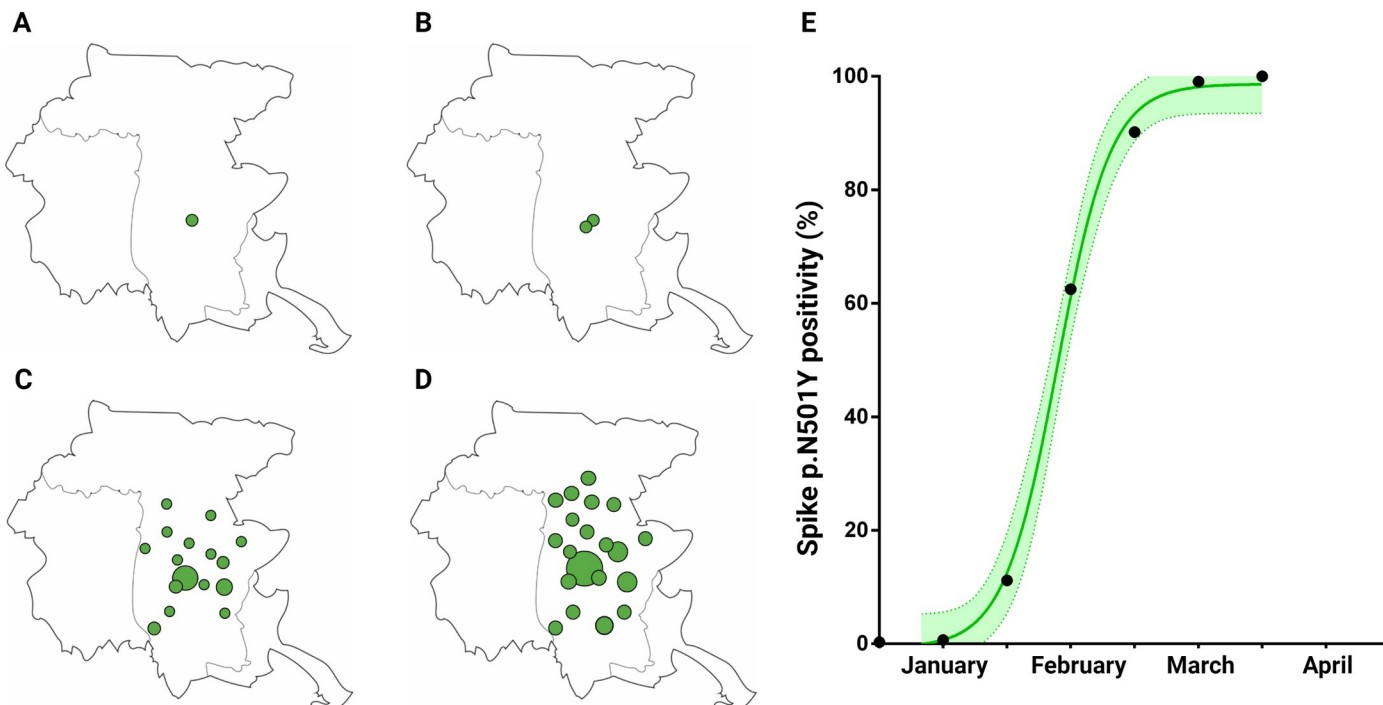

**Fig 1. Expansion of Spike p.N501Y positive samples in Friuli Venezia Giulia.** A-D, p.N501Y spread in the province of Udine. Green dots represent p.N501Y positive samples from January 1st to January 15th (A), from January 16th to January 31st (B), from February 1st to February 15th (C), from February 16th to February 28th (D). E, relative abundance of the p.N501Y positive samples in about 15 weeks. Dots represent the estimated frequency in each fortnight. Dotted lines show 95% confidence interval. Created with Biorender.com.

**Table 1. Retrospective distribution of SARS-CoV-2 Spike p.N501Y positive samples in our cohort.**

| Time interval | Samples analyzed (N) | p.N501Y positive (N) | Percentage (%) |
|---|---|---|---|
| January 1st—January 15th | 288 | 1 | 0.3 |
| January 16th–January 31st | 304 | 2 | 0.7 |
| February 1st–February 15th | 367 | 41 | 11.2 |
| February 16th–February 28th | 152 | 95 | 62.5 |
| March 1st -March 15th | 174 | 157 | 90.2 |
| April 1st -April 15th | 223 | 221 | 99.1 |

After quality filtering, 479 sequences were retained for further analysis. The mean coverage was 5158 X [95% CI: 4925–5392], with a mean uniformity of sequencing of 94.14% [95% CI: 93.72–94.55]. Indeed, all of the SARS-CoV-2 B.1.1.7 genomes analyzed in our cohort contained all the so-called signature mutations in the Spike glycoprotein, including p.H69-V70del, p.Y144del, p.N501Y, p.A570D, p.P681H, p.T716I, p.S982A, and p.D1118H [3]. Noteworthy, two specimens collected in mid-February turned out to possess the p.E484K but not p.N501Y. After pangolin web app annotation both samples turned out to belong to the B.1.1.318 lineage. Indeed, it was defined as a variant of interest (VUI202102/04) in February 2021 and, since April 2021, 536 cases have been defined worldwide, with a majority in the US, UK, and Germany (https://cov-lineages.org/lineages/lineage_B.1.1.318.html). To get a more insight in the distribution of lineages within our territory, a phylogenetic analysis of the 479 sequences has been performed (Fig 2, panel A). Relative abundance of each lineage is summarized in Fig 2, panel B. Eventually, our 479 sequences have been analyzed with the Nextclade online tool in order to map them within the SARS-CoV-2 global landscape (Fig 2, panel C).

Subsequently, samples have been analyzed in order to putatively pinpoint whether there was any enrichment in non-synonymous mutations rather than the B.1.1.7-defining ones. 39% of sequences harbored NSP12b p.P218L (Orf1ab p.P4619S), NSP14 p.P451S (Orf1ab p. P6376S), Orf6 p.D61L and Orf8 p.K68* mutations. 20% of viral genomes analyzed presented the NSP4 p.L206F (Orf1ab p.L2969F) in combination with N p.A156S. 14.5% presented the combination of NSP3 p.E405A (Orf1ab p.E1223A) and Orf7a p.V106L. Among these mutations, only the NSP12b p.P218L is already indexed on the CovidMiner data portal with a frequency of 16% among the deposited sequences. Furthermore, the Nucleocapsid p.A156S substitution has been already discussed in the study of Rahman and colleagues, highlighting a mutation-induced shift in protein stability with a decrease in flexibility. Indeed, it has been demonstrated that A156 is among the critical residues that interacts with diverse drug candidates (i.e., Conivaptan, Ergotamine, Venetoclax, Rifapentine), meant to interfere with Nucleocapsid-related functions.

We, then, focused our attention of mutations occurring on the Spike glycoprotein. In the first four time points, we only detected two SNPs that differs from the B.1.1.7-defining ones: the p.D138H substitution (N = 3) and the p.D796H (N = 1). Analyzing the last two time points, in concurrence with the exponential growth in B.1.1.7 lineage frequency, a higher number of Spike mutations have been uncovered. Table 2 and Fig 3 summarize our findings.

It should be noticed that most of these substitutions are rare mutations, highlighted in a single specimen. This prevents us from making any epidemiological or clinical conclusions on the putative consequence of these variants.

## Discussion

The emergence of diverse variants of concern in the ongoing SARS-CoV-2 pandemic requires rapid genomic, epidemiological, and clinical characterization to convey public health

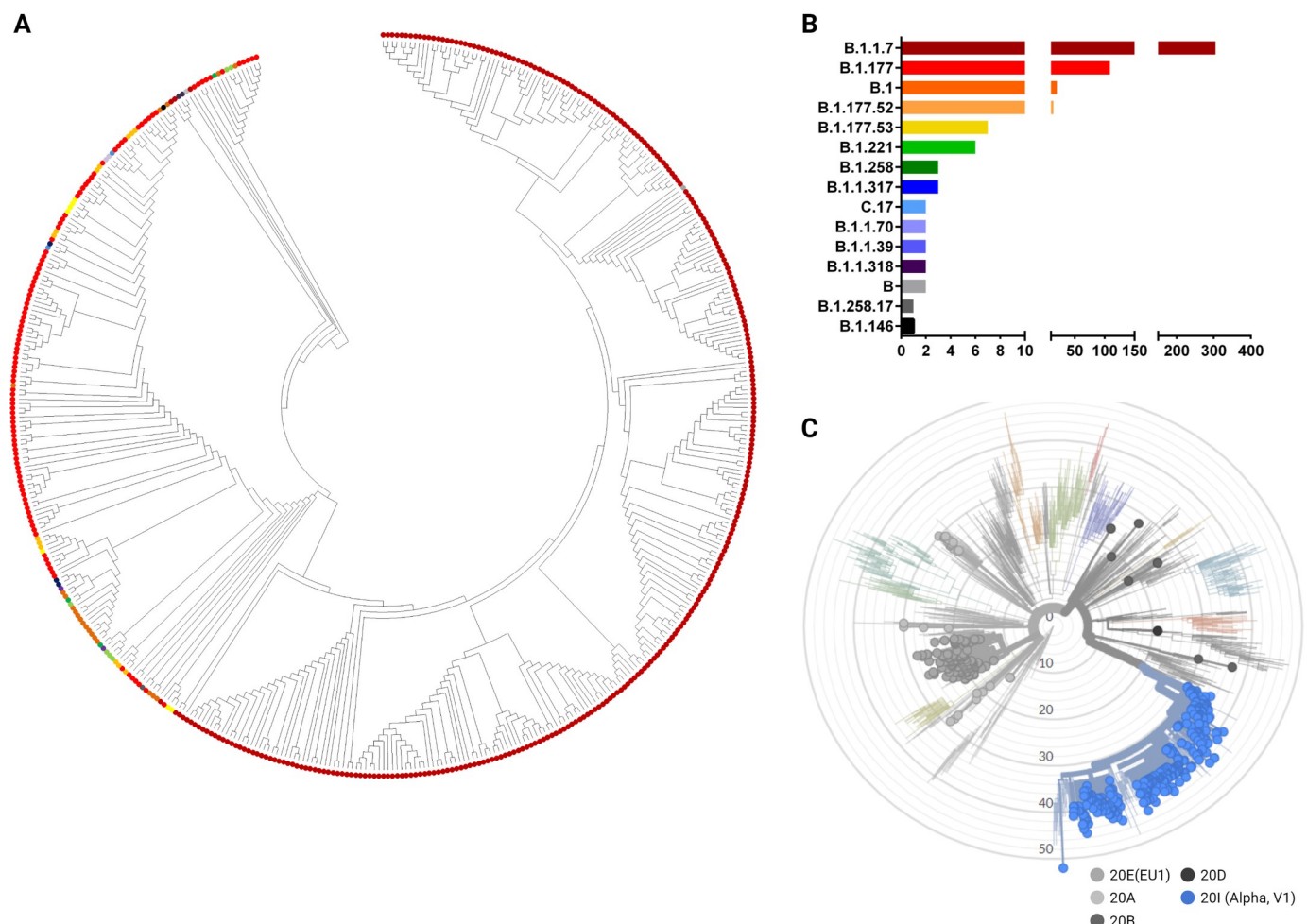

**Fig 2. Phylogenetic analysis and clades distribution of positive samples.** A, phylogenetic tree of 479 sequences based on the maximum likelihood method of MEGA v.11. B, distribution of PANGO lineages between January 1st and April 15th in Friuli Venezia Giulia. C, clades distribution according to the Nextclade online tool. Panel A and B share the same color code. Created with Biorender.com.

responses and reorganize hospitalization. We retrospectively investigated the presence of the three previously reported VOC (B.1.1.7, B.1.351 and P1) in a cohort of 1508 specimens, found positive for SARS-CoV-2 infection in a time span of about 4 months in the northeastern area of the Italian peninsula. 479 samples were, then, randomly selected to undergo NGS. In this work, we chose to apply a primary HRM-based screening, followed by the analysis of only a third of samples by NGS. Surely the greatest benefit in using HRM is the possibility to process a large number of samples at a minimum cost and in a very short time, which is not possible using NGS. The latter, in fact, although it gives more informative results in terms of classification of viral sequences and in the discovery of novel mutations and new lineages, is time consuming and requires a huge economic and technological effort that makes it difficult to be applied for the timely tracking of VOC/VUI-infected patients. In our opinion, the combined use of these two techniques is a good compromise that has the ultimate goal of focusing efforts on samples that deviate from the endemic lineages of a certain area in a specific timeframe.

From the beginning of the pandemic, several containment measures were taken from the Italian Government. In particular, during late 2020, Italy was divided into three major risk-zones, calculated by the Italian technical scientific committee taking in account several

**Table 2. Novel spike mutation detected in the B.1.1.7 lineage genome background.**

| Spike mutation | Number of genomes | Occurrence in the cohort | Percentage in CovidMiner |
|---|---|---|---|
| p.L5F | 4 | 1.06% | 2.20% |
| p.V11F | 1 | 0.26% | n.a. |
| p.L18F | 1 | 0.26% | 7.50% |
| p.T19I | 1 | 0.26% | n.a. |
| p.P26L | 1 | 0.26% | 0.10% |
| p.A67V | 1 | 0.26% | n.a. |
| p.I68_H69del | 1 | 0.26% | n.a. |
| p.V70F | 1 | 0.26% | n.a. |
| p.D138H | 3 | 0.79% | 0.20% |
| p.D138_Y145del | 1 | 0.26% | n.a. |
| p.D215Y | 1 | 0.26% | n.a. |
| p.P330S | 1 | 0.26% | n.a. |
| p.P384S | 4 | 1.06% | n.a. |
| p.E484K | 2 | 0.53% | 1.50% |
| p.E583D | 3 | 0.79% | 0.30% |
| p.V608I | 1 | 0.26% | n.a. |
| p.N658K | 1 | 0.26% | n.a. |
| p.A694V | 1 | 0.26% | n.a. |
| p.N751S | 1 | 0.26% | n.a. |
| p.V772I | 1 | 0.26% | 0.50% |
| p.L922F | 1 | 0.26% | n.a. |
| p.V1065L | 2 | 0.53% | n.a. |
| p.P1079S | 1 | 0.26% | n.a. |
| p.H1101Y | 1 | 0.26% | n.a. |
| p.V1104L | 1 | 0.26% | n.a. |
| p.D1153H | 1 | 0.26% | n.a. |
| p.V1264L | 1 | 0.26% | 0.10% |

monitoring objective parameters (i.e., reproduction number (R) and growth rate (r)). Concurrently, in order to minimize risks, the whole nation was considered as a 'high risk zone'; long-ranging transfers and all private gatherings involving more than two persons from different families were banned by law [14]. In January 2021, a gradual mitigation of pandemic-related containment measures, and the consequent opening of the boarders, was authorized.

In this study, we showed that the B.1.1.7 lineage rapidly spread becoming the prevalent lineage from January to March 2021. Indeed, a progressive increase in positive cases of SARS-CoV-2 infections and the concomitant emergence of the Alpha variant were reported in diverse regions of Italy [6, 7, 15, 16]. Lai and colleagues reported that the B.1.1.7 lineage significantly increased from 3.5% in December 2020 to 86.7% in March 2021, replacing the previously circulating variant B.1.177 [7]. Moreover, they assessed that the most important increase in its prevalence was observed from the second week of January to the end of the same month, reaching 73.7% prevalence at the end of February. Our data perfectly fits in this overview.

Thus, it is tempting to speculate that the relaxation of pandemic-related containment measures and the opening of international travels may be implicated in the emergence and subsequent spread of the Alpha variant in our region.

Moreover, our findings are consistent with those gathered worldwide [17] where this lineage established itself as the dominant one within a few months after its detection. Indeed, it has been shown that the p.N501Y mutation is associated to an augmented affinity to the

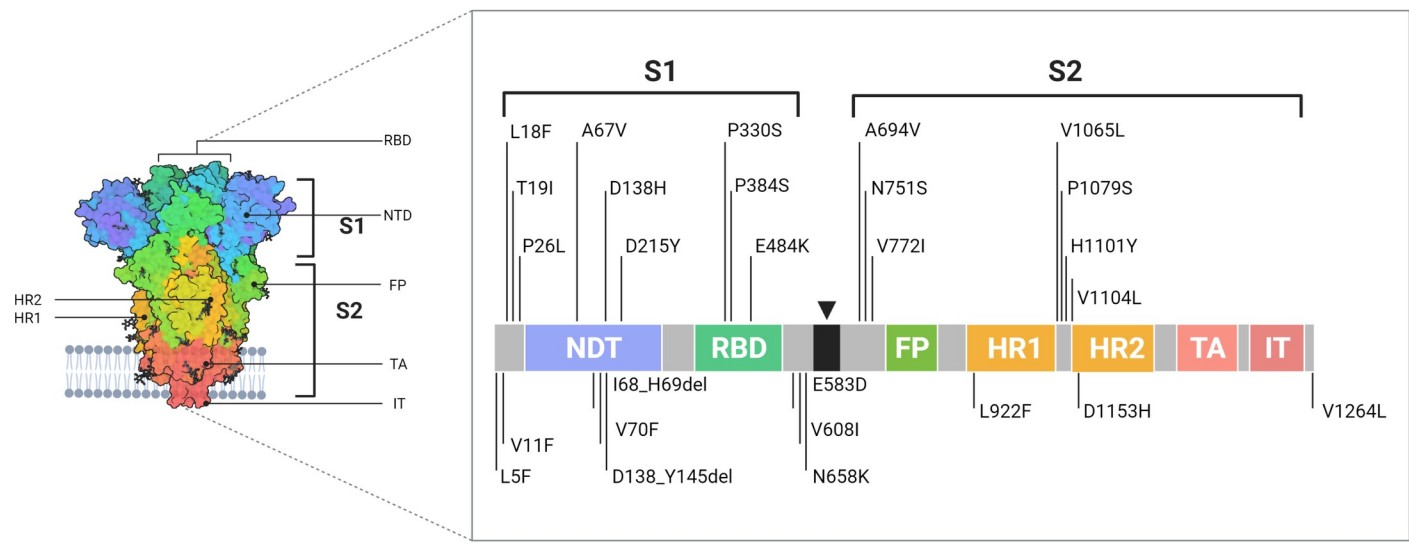

**Fig 3. Schematic representation of the Spike glycoprotein and the rare mutations highlighted by NGS.** Cartoon describing the SARS-CoV-2 Spike glycoprotein 3D structure and its interaction with the cellular plasma membrane together with a graphical representation of Spike-related sequencing data. RBD: RNA binding domain; NTD: N-terminal domain; FP: fusion protein; TA: transmembrane anchor; IT: intracellular tail; S1: subunit 1; S2: subunit 2; HR1: heptad repeat region 1; HR2: heptad repeat region 2; the inverted black triangle represents the amino acid bridge. Created with Biorender.com.

angiotensin-converting enzyme 2 (ACE2) receptor [18] and an increased transmissibility compared to previous circulating SARS-CoV-2 lineages [19]. Notwithstanding, no strong evidences have been made on an increased clinical severity or immune escape capability in patients infected by the B.1.1.7 lineage [20–22]. Indeed, epidemiological data from the former province of Udine highlighted a rapid increase in infection and related hospitalizations after the stabilization of the B.1.1.7 lineage within the territory (Fig 4).

Once reached the plateau (i.e., > 90% prevalence), our data showed the onset of several non-canonical mutations on the B.1.1.7-related Spike sequence, a possible hint of the fact that the viral genome is still in continuous evolution. Of particular concern is the B.1.1.7 lineage harboring the p.E484K, since it has been demonstrated that it enhances escape from neutralizing antibody inhibition *in vitro*, and reduces efficacy of the vaccine [23]. Nonetheless, the presence of the p.E484K mutation in the B.1.1.7 background has been recently reported, named VOC-21FEB-02 in late March 2021. p.E484K is a mutation of concern with regards to antigenic change and receptor binding avidity; it is potentially more concerning when combined with N501Y. It has been first identified in mid-February and it has been postulated to be selected by convalescent and vaccine-derived antisera.

Indeed, mRNA based (Moderna, Pfizer-BioNTech), adenoviral based (Vaxzevria, Johnson&Johnson), and protein-based based (Novavax) vaccines are only able to induce a monoclonal response towards the Asp614Gly bearing SARS-CoV-2 Spike protein, therefore a steady acquisition of mutations that could eventually lead to an enhanced fitness should be carefully monitored.

## Conclusions

Despite limitations, we were able to depict the geographic spread and genomic evolution of the SARS-CoV-2 B.1.1.7 lineage in our territory. Our data are a proof of concept that increased genomic surveillance will be crucial for detection of emerging variants of concern, which could escape natural- and vaccine-induced immunity, and for the coordination of appropriate health measures to contain the infection.

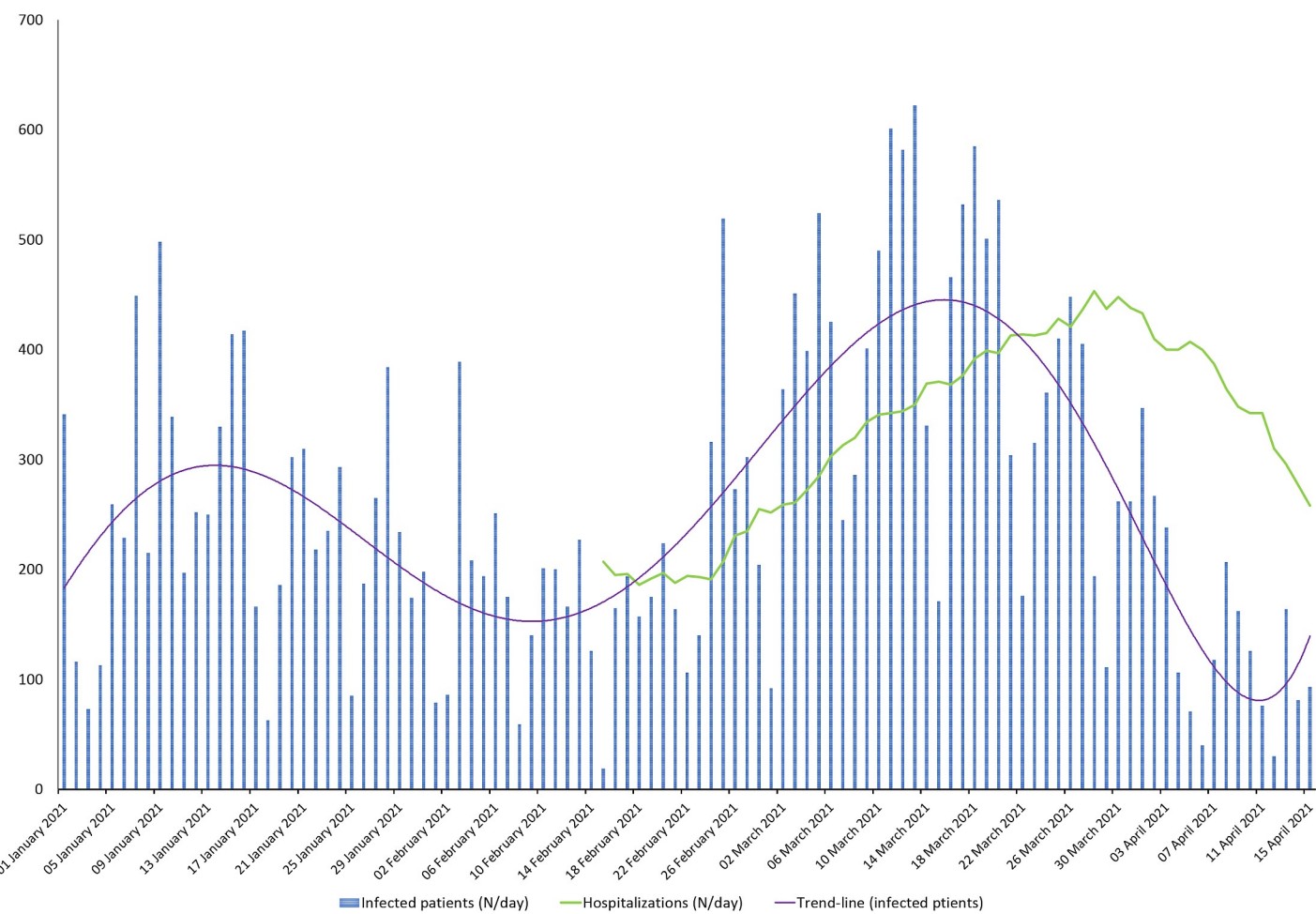

**Fig 4. Epidemic curve of confirmed SARS-CoV-2 infected patients in Udine by day.** Blue bars represent the number of samples tested positive for SARS-CoV-2 infection per day between January 1st to April 15th, the period covered by our analysis. The green line represents the number of hospitalized patients due to COVID-19 per day from mid-February. The purple line represents the trend line of daily infections.

## Supporting information

**S1 Table. Approximate correlation between the major SARS-CoV-2 PANGO lineage, GISAID Nextstrain clade and WHO nomenclature.**
(DOCX)

**S2 Table. Accession ID and virus names of the 479 sequences uploaded in the GISAID database.**
(DOCX)

## Acknowledgments

Sequences have been shared in the GISAID database (https://www.gisaid.org/).

## Author Contributions

**Conceptualization:** Catia Mio, Chiara Dal Secco, Stefania Marzinotto.

**Data curation:** Catia Mio, Chiara Dal Secco.

**Formal analysis:** Catia Mio, Chiara Dal Secco, Emanuela Sozio.

**Investigation:** Catia Mio, Chiara Dal Secco, Claudio Bruno, Santa Pimpo, Elena Betto, Martina Bertoni, Emanuela Sozio.

**Methodology:** Catia Mio, Chiara Dal Secco, Stefania Marzinotto.

**Project administration:** Corrado Pipan, Carlo Tascini, Francesco Curcio.

**Supervision:** Stefania Marzinotto, Corrado Pipan, Carlo Tascini, Giuseppe Damante, Francesco Curcio.

**Writing – original draft:** Catia Mio, Chiara Dal Secco.

**Writing – review & editing:** Stefania Marzinotto, Claudio Bruno, Santa Pimpo, Elena Betto, Martina Bertoni, Corrado Pipan, Emanuela Sozio, Carlo Tascini, Giuseppe Damante, Francesco Curcio.

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
