## [Decision Letter · Decision Letter 0]

7 Sep 2021

PONE-D-21-25721Local occurrence and fast spread of B.1.1.7 lineage: a glimpse into Friuli Venezia GiuliaPLOS ONE

Dear Dr. Curcio,

Thank you for submitting your manuscript to PLOS ONE. After careful consideration, we feel that it has merit but does not fully meet PLOS ONE’s publication criteria as it currently stands. Therefore, we invite you to submit a revised version of the manuscript that addresses the points raised during the review process.

Ma conclusion = major revision and The results are of interest as indicated by the reviewerw but some details are lacking. First of all, please made all data underlying the findings in the manuscript fully available. Ie list the sequence accession number and specific list. Explain what is the reference sequence in full and please answer to all the detailled question from reviewers. Please note that specific comment from reviewer 1 is in attachement.

We look forward to receiving your revised manuscript.

Kind regards,

Pierre Roques, Ph.D.

Academic Editor

PLOS ONE

Journal Requirements:

2. You indicated that you had ethical approval for your study. In your Methods section, please ensure you have also stated whether you obtained consent from participants included in the study or whether the research ethics committee or IRB specifically waived the need for their consent.

Reviewers' comments:

Reviewer's Responses to Questions

**Comments to the Author**

1. Is the manuscript technically sound, and do the data support the conclusions?

Reviewer #1: Yes

Reviewer #2: Partly

Reviewer #3: Yes

2. Has the statistical analysis been performed appropriately and rigorously? 

Reviewer #1: I Don't Know

Reviewer #2: Yes

Reviewer #3: Yes

3. Have the authors made all data underlying the findings in their manuscript fully available?

Reviewer #1: No

Reviewer #2: No

Reviewer #3: No

4. Is the manuscript presented in an intelligible fashion and written in standard English?

Reviewer #1: Yes

Reviewer #2: No

Reviewer #3: Yes

5. Review Comments to the Author

Reviewer #1: Overview of the manuscript’s

On real-time screening is the cornerstone strategy to fight against current Covid pandemic, especially variants sequencing. Variants increase transmissibility, virulence. Their association to fatal diseases progression justifies the relationship between variants progression and the hospitalization cases number in this paper. Is the detection of new mutations important, here? Giving more details in the clinical signs of patients (with B.1.1.7 detection), in lineages diversity in this region and at frontiers should be benefices to paper.

Confer attachments to recommendation

Reviewer #2: The paper by Mio et al. describes the emergence and spread of the SARS-CoV-2 alpha variant between January and April 2021 in the Udine region of Italy. The presence of the alpha variant was assessed by HRM as well as NGS, however, in its current form it is in need of a major review, before being suitable for publication.

Comments:

1. Was the typing of all samples only based on the HRM (i.e. presence/absence of these two mutations) or were they all confirmed by NGS?

2. Are there other strains that possess these two mutations but do not belong to the alpha variant?

3. The NGS part is not presented in a clear manner. How many samples were analysed? What were the criteria for choosing these samples?

4. I would suggest to add a phylogenetic analysis for the sequences obtained including sequences of surrounding regions during the same time period. Also, the accession numbers are missing.

5. Is clinical data available for this time period? It would be interesting to evaluate the impact of the spread of the alpha variant with regard to hospitalisations, severity of disease course and mortality.

6. The discussion needs expansion. What happened in this time period in other parts of Italy (or Europe) with regard to the alpha variant?

7. In many parts I had to struggle with the language; I am not sure what “private” mutations refer to or how samples were “harvested”. I strongly suggest that the manuscript is proof-read by a native English speaker before resubmission

8. Table 1 should be moved to supplementary information; the WHO classification should be indicated in a separate column instead of in brackets in the Nextstrain clade column

Reviewer #3: The manuscript from Mio et al. retrospectively investigated the local occurrence and spread of B.1.1.7 in a region in Northeastern Italy. The information provided in this manuscript can be useful for understanding the local pandemic development. However, some detailed information should be added. I would like to suggest revisions regarding the topics below:

1. About data availability:

The authors describe “sequences are available from the GISAID database”, without providing detailed information. For the 1508 sequences, the authors should provide GISAID Accession ID, so anyone who read the paper could directly check the sequences.

2. In Table 1, the authors summarised mutation information for several variants. The authors did not describe which reference genome they refer to when discussing mutations. The “reference genome” information should be provided, because the mutations can be different if a different reference genome is used in the analysis.

3. In Table 2, information about the time interval “March 16th - March 31th” is missing. Any specific reason for that? Otherwise, the authors should provide relevant information for this time period as well.

6. PLOS authors have the option to publish the peer review history of their article (what does this mean?). If published, this will include your full peer review and any attached files.

Reviewer #1: No

Reviewer #2: **Yes: **Jan Richter

Reviewer #3: No

---

## [Author Response · Author response to Decision Letter 0]

22 Oct 2021

Point-by-point answers to the reviewer’ requests:

Reviewer #1: 

Comments to the Author:

Overview of the manuscript’s : On real-time screening is the cornerstone strategy to fight against current Covid pandemic, especially variants sequencing. Variants increase transmissibility, virulence. Their association to fatal diseases progression justifies the relationship between variants progression and the hospitalization cases number in this paper. Is the detection of new mutations important, here? Giving more details in the clinical signs of patients (with B.1.1.7 detection), in lineages diversity in this region and at frontiers should be benefices to paper.

Summary of the research

The researcher group takes the opportunity to combine High Resolution Melting (HRM) approach and Next Generation Sequencing analysis (NGS) to screen the first four months of year and bring an overview to virus genomic evolution from January 1st 2021, until April 15th 2021 in this particular north-eastern region of Italy. The conclusion is an increasing of variant for this period as well as cluster sub-lineages with interest concern B.1.1.318 lineage.

Keys list:

Strengths

Sequenced samples number

Screening methods combination (HRM approach and sequencing)

Ethical guidelines respect

Spatio-temporal information of reported cases

Weaknesses:

Short duration of study (4 months)

Unuseful work in new mutation detection

No clinical signs data and demographic information of cases

No advices and perspectives to push interest of this paper

Recommended course of action (major):

1. Introduction.

People speak about 19 A 20 A clades (line 40-41) without referring to Nextstrain nomenclature and its references (add website link and version).

The website and the relative version have been added to the revised version of the manuscript (page 3, line 45).

2. In addition, it needs to recall that Pangolin nomenclature system is not a way to “solve a problem” to GISAID or Nextstrain assignement supports. Pangolin is a complementary approach to other, and in especially to focus on the local dynamic and through lineages diversity instead of clades. Conjugating nomenclature systems are so interesting method. Don't forgot also WHOs' (Alpha, Beta, etc..). It needs to be mentioned in the introduction too.

The paragraph discussing the nomenclature system has been rewritten, including the WHO-related classification (page 3, lines 47-56). 

3. Epidemiological context of pandemic in Italy and in this specific region prior of the study should be mentioned at this time point. This is to better understand the possible consequence in variants emergence and paper' interest.

We thank the reviewer for the suggestion. Retrospective studies highlighted that in Italy the main circulating lineage were B.1 and B.1.177 together with their sub-lineages, representing about 70% of cases in late 2020. In our region, the prevalent lineages were the B.1.177 and B.1.258 ones, according to the sequences related to November-December 2020 uploaded in the GISAID database. This information has been added to the text (page 7, lines 61-63) together with appropriate references. 

4. Material and methods

Precise the process of samples selection for 4 months: random n samples by months, no rule and only available samples?

Samples were randomly selected from all the SARS-CoV-2 swabs tested positive in the timeframe analysed in this study. This sentence has been added in the new version of the revised manuscript (page 7, line 77-84).

5. Add demographical and clinical information related to sequencing samples

We thank the reviewer for the suggestions to improve our manuscript. Unfortunately, we are not in possession of the cohort-related clinical data. 

6. Give mor info in High resolution melting (HRM) technic and its benefices: melting temperature analysis...

Thermal shifts have been added to the revised version of the manuscript, to better comprehend this method (page 8, lines 97-98).

7. Notice date of assignement by using Nextstrain and Pangolin (line 101 103) due to update in Continuous

Databases and tools versions have been included in the material and methods section (page 9, lines 121-129).

8. Results

Lines 108- 115 need to be moved to in material ad methods

We thank the reviewer for the suggestions to improve our manuscript. The aforementioned sentences have been edited and partially added to the material and methods section (page 7, lines 77-84).

9. Data shows only lineages reports (Pangolin assignement). Its of Nextstrain assignement miss and added concerning 1508 seq (i.e.: is it clade 20Ia and for all?)

Since PANGO nomenclature turned out to be the most widely used (https://doi.org/10.1038/s41576-021-00408-x), before the WHO reclassification, which is indeed posterior to the timeframe described in this paper, we preferred to use one nomenclature only. Notwithstanding, PANGO lineages and Nextsrain clades have been summarized in Figure 2.

10. Figure 2: most of lineage are B.1.1.7 lineages (major %). but what about the % remaining?

Data related to lineage distribution are now summarized in Figure 2. 

11. Figure 1: It shows delay between infected patients (blue line) and hospitalizations number (green line). In addition, it should be possible to link the increasing in variants incidence to specific clinical signs changes and/or demographical population targeted (ages, sex etc..) for 4 months at hospital (i.e: table with progression index, age, sex, etc...). It provides opportunity to perform additionnal multivariable statistical analysis in correlation between to these events.

We thank the reviewer for the suggestions to improve our manuscript. Unfortunately, we are not in possession of the cohort-related clinical data.

12. In contrast, work in new mutations detection has not to be place on main document. It makes confusing analysis without arguing in interested to support main message. This part may be rewrited by only speaking in known mutations concern to variant leading to review figure 1 C/D and Table 2 to suppl.

We thank the reviewer for the suggestions to improve our manuscript. Since the continuous monitoring in the establishment of novel mutation is the key to overcome the occurrence of future variants of concern, we believe that tracking novel and rare mutation is crucial in the pandemic outbreak.

13. Discussion 

Arguing in benefices to combine HRM + sequencing approaches, or oppose to it, as a timeconsuming as well as too restrictive (no new variants discovery) approach compared to directly perform sequencing of all positive samples. NGS allows to open the opportunity to detect other variants here, already described in worldwide and to frontiers as well as to propose under monitored variants as concern. Moreover, it is surprising to only detect 2 lineages B.1.1.7 and B.1.1.318. In general, SARS-Cov-2 sequencing in other countries shows diversity in lineages more and even in clades. This poorness is it justify or not (among >1500 seq.)? No detection of parental lineages (i.e: B.1) ?

We implemented the Discussion section arguing pro and cons of the approach used in this manuscript (Pages 15-16, lines 238-247). Moreover, a better description on lineage heterogeneity in sequenced samples together with a representative figure have been added to the text.

14. Speak in possible causes of variant introduction at the beginning of year by providing information to reported cases and/or to frontiers regions (business exchanges, traveler, unlock-down event, vacancy…).

Since we do not possess epidemiological data related to our cohort, we prefer not to speculate on the possible way in which the alpha variant has established itself in our territory.

15. In line 50, people mention “Viral genome sequencing coupled to epidemiological surveillance is mandatory”, it is a pity that the sequencing is not associated to epidemiological during the study period to discuss on it here.

Unfortunately, we do not possess these data.

16. Conclusion 

No advice and/or perspectives to use sequencing at hospital (to management) as well at frontiers to target new variant introductions in this strategic part of the Italian region.

Considerations of using an NGS-based screening protocol were included in the discussion section of our manuscript. NGS, indeed gives more informative results in terms of classification of viral sequences and in the discovery of novel mutations and new lineages. Nonetheless, it is time consuming (72 hours are needed to obtain lineage information) and it requires a huge economic and technological effort that makes it difficult to be applied for the timely tracking of VOC/VUI-infected patients.

17. Graphics 

Table 1: this relationship between GISAIS/Nextstrain and Pangolin nomenclature systems is well described in Alma et al, (https://doi.org/10.2807/1560-7917.ES.2020.25.32.2001410). Paper could be noticed to remove table 1. Or table1 should be redesigned (Column 1 GISAID, Column 2 Nextstrain, Colum 3 Pangolin) and put on suppl. In addition, table 1 needs more legend details (choice of some variants and not others, mutations code (p.?) and also mentioned WHO nomenclature (alpha, beta etc..).

The aforementioned paper has been added to the manuscript text and Table 1 has been moved to Supplementary Data.

18. Figure 1 1A: 4 maps, one for each month, allow to localize positive cases in space (in green), identify some of them associated to variant (in black) and possibly show cluster emergence in time (i.e. radiation or centralization). 1B: it is related to table 3 and better to put together one above other. 1C and 1D are structural and mutations concern: to put apart in a new figure, especially to suppl. as already recommended.

We thank the reviewer for the suggestions to improve our manuscript. Figures have been edited to better describe our data.

Recommended course of action (minor):

1. Introduction it may be important to justify the neutral genetic drift (line 33) and enhance of viral fitness affect in mutations constellations (line 53) with SARS-Cov-2 references.

Appropriate references have been added to the main text.

2. Methods 

% of targeted coverage, number of sequencing with >10% missing genome (no success).

Data relative to NGS-based data analysis have been added to the main text. 

3. Results 

Bring information in patient treatments and management results to variants emergence at hospital. It might affect the quality of diagnostic and/or sequencing success?!

Unfortunately, we do not possess these data.

4. Discussion 

The emergence of variant is it associate to guidelines changes in local regions such as lockdown, diagnostic or vaccine recommendation?

From the beginning of the pandemic, several containment measures were taken from the Italian Government. In particular, during late 2020, Italy was divided into three major risk-zones, calculated by the Italian technical scientific committee taking in account several monitoring objective parameters (i.e., reproduction number (R) and growth rate (r)). Concurrently, in order to minimize risks, the whole nation was considered as a 'high risk zone'; long-ranging transfers and all private gatherings involving more than two persons from different families were banned by law. In January 2021, a gradual mitigation of pandemic-related containment measures, and the consequent opening of the boarders, was authorized. Indeed, a progressive increase in positive cases of SARS-CoV-2 infections and the concomitant emergence of the Alpha variant were reported. Thus, it is tempting to speculate that the relaxation of pandemic-related containment measures and the opening of international travels may be implicated in the emergence and subsequent spread of the Alpha variant in our region. This information has been added to the text (page 16, lines 250-266) together with appropriate references. 

5. Remove 188-195 lines as losing the lector to main goal

We thank the reviewer for the suggestion. Notwithstanding, we believe that the continuous monitoring and recording of new rare mutations is mandatory to comprehensively investigate and keep track of virus evolution and could shed light on possible causes of reduced vaccine efficacy. Therefore, we prefer to keep this section in our manuscript. 

6. Figure Improve the resolution quality of figures

As described in the journal guidelines, higher resolution images will be introduced if the manuscript is accepted for publication.

Reviewer #2

Comments to the Author:

The paper by Mio et al. describes the emergence and spread of the SARS-CoV-2 alpha variant between January and April 2021 in the Udine region of Italy. The presence of the alpha variant was assessed by HRM as well as NGS, however, in its current form it is in need of a major review, before being suitable for publication.

Comments:

1. Was the typing of all samples only based on the HRM (i.e. presence/absence of these two mutations) or were they all confirmed by NGS?

1508 samples were screened by HRM and then randomly selected for sequencing. After quality-based filtering, 479 sequences were retained for further analysis (Page 8, line 101). A more detailed workflow has been added in the new version of the manuscript.

2. Are there other strains that possess these two mutations but do not belong to the alpha variant?

The analysis of the sequencing data did not reveal other strains possessing both p.E484K and p.N501Y other than the alpha lineage. A summary of the non-alpha lineages highlighted in our study has been included in the new version of the manuscript (Figure 2).

3. The NGS part is not presented in a clear manner. How many samples were analysed? What were the criteria for choosing these samples?

We thank the reviewer for the suggestions to improve our manuscript. A better description of the NGS-based workflow has been added in the material and methods session. Samples underwent HRM screening, were randomly selected for NGS. After filtering criteria based on coverage (FDP>500), percentage of gaps (N< 20% of the entire sequence), 479 sequences were retained for further analyses.

4. I would suggest to add a phylogenetic analysis for the sequences obtained including sequences of surrounding regions during the same time period. Also, the accession numbers are missing.

A phylogenetic analysis of the 479 sequences included in our manuscript has been added (Figure 2). The list of the GISAID Accession ID is now available as Supplementary Table 2.

5. Is clinical data available for this time period? It would be interesting to evaluate the impact of the spread of the alpha variant with regard to hospitalisations, severity of disease course and mortality.

We thank the reviewer for the suggestions to improve our manuscript. Unfortunately, do not possess the cohort-related clinical data. 

6. The discussion needs expansion. What happened in this time period in other parts of Italy (or Europe) with regard to the alpha variant?

We thank the reviewer for the suggestion. Retrospective studies highlighted that in late 2020 in Italy the main circulating lineage were B.1 and B.1.177 together with their sub-lineages, representing about 70% of cases. From December 2020, a progressive increase in positive cases of Alpha-related infections were reported in different part of Italy, with a trend overlapping the one presented in our manuscript. Indeed, Lai and colleagues reported that the B.1.1.7 lineage significantly increased from 3.5% in December 2020 to 86.7% in March 2021 in diverse Italian regions [doi: 10.1186/s12985-021-01638-5]. Moreover, they assessed that the most important increase in its prevalence was observed from the second week of January to the end of the same month, reaching 73.7% prevalence at the end of February. Our data perfectly fits in this overview. This information has been added to the text (page 16, lines 250-266) together with appropriate references. 

7. In many parts I had to struggle with the language; I am not sure what “private” mutations refer to or how samples were “harvested”. I strongly suggest that the manuscript is proof-read by a native English speaker before resubmission.

In genetics, “private mutations” are variations found usually only in a single family or a small population. Notwithstanding, the description of rare SNP identified in our cohort has been rewritten, to avoid any possible confusion. Moreover, the new version of the manuscript has been entirely revised for correct use of language and for typos.

8. Table 1 should be moved to supplementary information; the WHO classification should be indicated in a separate column instead of in brackets in the Nextstrain clade column.

Table 1 has been modified and moved to supplementary data.

Reviewer #3

Comments to the Author:

The manuscript from Mio et al. retrospectively investigated the local occurrence and spread of B.1.1.7 in a region in Northeastern Italy. The information provided in this manuscript can be useful for understanding the local pandemic development. However, some detailed information should be added. I would like to suggest revisions regarding the topics below:

1. About data availability: The authors describe “sequences are available from the GISAID database”, without providing detailed information. For the 1508 sequences, the authors should provide GISAID Accession ID, so anyone who read the paper could directly check the sequences.

We thank the reviewer for the suggestion. 1508 samples were screened by HRM and then randomly selected for sequencing. A more detailed workflow has been added in the new version of the manuscript. About 479 sequences were uploaded in the GISAID database. The list of the GISAID Accession ID is now available as Supplementary Table 2.

2. In Table 1, the authors summarised mutation information for several variants. The authors did not describe which reference genome they refer to when discussing mutations. The “reference genome” information should be provided, because the mutations can be different if a different reference genome is used in the analysis.

We thank the reviewer for the suggestion. The reference genome used in this manuscript is the NC_045512.2 assembly. This information has been added in the material and methods session (Page 8, line 105).

3. In Table 2, information about the time interval “March 16th - March 31th” is missing. Any specific reason for that? Otherwise, the authors should provide relevant information for this time period as well.

Samples related to the aforementioned time point have not been analyzed and therefore this interval is missing in Table 2.

---

## [Decision Letter · Decision Letter 1]

25 Nov 2021

Local occurrence and fast spread of B.1.1.7 lineage: a glimpse into Friuli Venezia Giulia

PONE-D-21-25721R1

Dear Dr. Curcio,

We’re pleased to inform you that your manuscript has been judged scientifically suitable for publication and will be formally accepted for publication once it meets all outstanding technical requirements.

Kind regards,

Pierre Roques, Ph.D.

Academic Editor

PLOS ONE

Additional Editor Comments (optional):

Reviewers' comments:

Reviewer's Responses to Questions

**Comments to the Author**

1. If the authors have adequately addressed your comments raised in a previous round of review and you feel that this manuscript is now acceptable for publication, you may indicate that here to bypass the “Comments to the Author” section, enter your conflict of interest statement in the “Confidential to Editor” section, and submit your "Accept" recommendation.

Reviewer #2: All comments have been addressed

Reviewer #3: All comments have been addressed

2. Is the manuscript technically sound, and do the data support the conclusions?

Reviewer #2: Yes

Reviewer #3: (No Response)

3. Has the statistical analysis been performed appropriately and rigorously? 

Reviewer #2: Yes

Reviewer #3: (No Response)

4. Have the authors made all data underlying the findings in their manuscript fully available?

Reviewer #2: Yes

Reviewer #3: (No Response)

5. Is the manuscript presented in an intelligible fashion and written in standard English?

Reviewer #2: Yes

Reviewer #3: (No Response)

6. Review Comments to the Author

Reviewer #2: During the review all points raised previously have been addressed in a satisfactory manner. The manuscript is now deemed acceptable for publication.

Reviewer #3: (No Response)

7. PLOS authors have the option to publish the peer review history of their article (what does this mean?). If published, this will include your full peer review and any attached files.

Reviewer #2: **Yes: **Jan Richter

Reviewer #3: No

---

## [Editor Report · Acceptance letter]

3 Dec 2021

PONE-D-21-25721R1 

Local occurrence and fast spread of B.1.1.7 lineage: a glimpse into Friuli Venezia Giulia 

Dear Dr. Curcio:

I'm pleased to inform you that your manuscript has been deemed suitable for publication in PLOS ONE. Congratulations! Your manuscript is now with our production department. 

Kind regards, 

on behalf of

Dr. Pierre Roques 

Academic Editor

PLOS ONE